# Circulating Tumor Cell Clusters: United We Stand Divided We Fall

**DOI:** 10.3390/ijms21072653

**Published:** 2020-04-10

**Authors:** Samuel Amintas, Aurélie Bedel, François Moreau-Gaudry, Julian Boutin, Louis Buscail, Jean-Philippe Merlio, Véronique Vendrely, Sandrine Dabernat, Etienne Buscail

**Affiliations:** 1INSERM U1035, Université de Bordeaux, 33000 Bordeaux, France; sam.amintas@gmail.com (S.A.); aurelie.bedel@u-bordeaux.fr (A.B.); francois.moreau-gaudry@u-bordeaux.fr (F.M.-G.); boutinjulian33@gmail.com (J.B.); jp.merlio@u-bordeaux.fr (J.-P.M.); veronique.vendrely@chu-bordeaux.fr (V.V.); 2Centre Hospitalier Universitaire (CHU) de Bordeaux, 33000 Bordeaux, France; 3Centre Hospitalier Universitaire (CHU) de Toulouse, 31000 Toulouse, France; buscail.l@chu-toulouse.fr (L.B.); ebuscail@me.com (E.B.); 4INSERM UMR 1037, Toulouse Centre for Cancer Research, University of Toulouse III, 31000 Toulouse, France; 5INSERM UMR-1220, IRSD University of Toulouse III, 31000 Toulouse, France

**Keywords:** liquid biopsy, circulating tumor cells clusters, circulating tumor cells, CTC, microemboli

## Abstract

The presence of circulating tumor cells (CTCs) and CTC clusters, also known as tumor microemboli, in biological fluids has long been described. Intensive research on single CTCs has made a significant contribution in understanding tumor invasion, metastasis tropism, and intra-tumor heterogeneity. Moreover, their being minimally invasive biomarkers has positioned them for diagnosis, prognosis, and recurrence monitoring tools. Initially, CTC clusters were out of focus, but major recent advances in the knowledge of their biogenesis and dissemination reposition them as critical actors in the pathophysiology of cancer, especially metastasis. Increasing evidence suggests that “united” CTCs, organized in clusters, resist better and carry stronger metastatic capacities than “divided” single CTCs. This review gathers recent insight on CTC cluster origin and dissemination. We will focus on their distinct molecular package necessary to resist multiple cell deaths that all circulating cells normally face. We will describe the molecular basis of their increased metastatic potential as compared to single CTCs. We will consider their clinical relevance as prognostic biomarkers. Finally, we will propose future directions for research and clinical applications in this promising topic in cancer.

## 1. Introduction

Since the first observation of circulating tumor cells (CTCs) in the mid-nineteenth century by Thomas Arsthworth, it took over a century to better characterize them because of many shortcomings, the major one being their rarity. About 10 years earlier, Virchow had detected tumor emboli entrapped in vasculature and proposed the first explanation of cancer dissemination by metastases. These clusters of cells were further described from the 1950s, with strong focus on their metastatic potential as compared to single circulating tumor cells (review in Reference [1]). There is already substantial literature on the molecular mechanisms linked with CTCs generation (review in Reference [2]), involving, for example, major cellular pathways effectors, like Erythroblastosis oncogene B (ERBB) family receptors [3]. On another side, mechanisms engaged in CTC clusters formation are still less developed. However, the dogma positioning single circulating cells as stemming metastasis based on their capacity to achieve epithelial to mesenchymal transition (EMT) (review in Reference [4]) did not propose CTC clusters as important actors of cancer dissemination. However, in depth investigations have revisited this hypothesis. By artificially modulating EMT, Beerling et al. showed the existence of epithelial-mesenchymal plasticity minimizing any differences in stemness between epithelial and mesenchymal states. This plasticity positions equally epithelial or mesenchymal circulating tumor cell to potential metastasis growth [5]. This is in agreement with the fact that no difference in EMT score was found in CTC clusters as compared to single CTCs originating from triple-negative breast cancer patient derived xenografts (PDXs) [6]. In the meantime, based on cytokeratin detection by immunocytochemistry or using magnetic beads, tumor cell clusters, mixed-cell doublets (one cytokeratin-positive and -negative cell), and mixed-cell clusters were identified, for example, in the blood of patients with colon cancer [7] or with prostate cancer [8], in the late 1990s, early 2000s. The most current CTC capture methods with antibodies detecting CTC surface proteins might be less efficient for CTC clusters as compared to single CTCs, possibly because antigens are largely masked by the CTC cluster nature itself. Thus, the real quantification of CTC clusters might be currently underestimated [9]. However, we will explore recent literature on CTC clusters and realize that their implication in cancer aggressiveness should not be neglected, as well as that they may even represent a valuable access to therapeutic optic. 

## 2. Origin and Dissemination of CTC Cluster

Although the past decade has seen intensification of CTC cluster characterization, their origin remains largely unknown. Among the possible mechanisms, cell jamming, or collective migration, is preferred to intravascular aggregation of single CTCs (Figure 1). This latter hypothesis was elegantly tested by tumor cell lineage tracing after engrafting equal mixes of cyan blue fluorescent protein CFP-expressing and tandem dimer TD-Tomato-expressing mammary tumor cells in the same mammary fat pad [10]. Authors found frequent polyclonal seeding of metastasis probably arising from oligoclonal CTC clusters. In addition, they observed no bicolored metastasis in the lungs after grafting the fluorescent tumor cells in distinct mammary fat pad of the same mouse or after injecting single fluorescent cells intravenously. In the same way, polyclonal metastasis, seeded by polyclonal CTC clusters, was a frequent event in murine pancreatic cancer [11]. Of note, both primary and metastatic tumors may contribute to release of CTC clusters, which may constitute local or distant tumor “self-seeding” sources [12]. Polyclonality of both the primary and the metastatic sites can arise from distinct tumor sub-clones seeding and differ based on the site of metastatic invasion [11]. This point is still debated, especially in light of the CD44 mediated homophilic aggregation of single CTC from a specific breast cancer line (BRX-50) and the dual deposition of fluorescent single cells in the same lung location after tail vein injection [13]. However, in this study, the authors did not clearly demonstrate the biclonal development of lung metastases.

Once the cluster has detached from the tumor mass, it has to enter blood vessels for hematogenous dissemination (Figure 1). It is now well-known that tumor vasculature is abnormal and offers CTCs and CTC clusters escape doors through leaky blood vessels [14]. Until recently, CTC clusters were believed to be unable to go through capillaries of 5–10 μm in diameter, especially when comprising numerous cells, and embolized and ruptured the vessel soon after their release. This rupture was also believed to contribute to metastatic dissemination. Surprisingly, CTC clusters traversed very narrow capillaries as a single file of still attached cells by their deformability properties and their adhesive strength [15], ruling out the assumption that CTC clusters are too large to reach distant organs and might get captured as microemboli in capillaries. This also explains that tumor-specific tropism of metastasis was early observed as not being always neighboring organs of the primary tumor but can also be distant [16]. Another way that CTC clusters may use to enter and also exit the blood vessels is using guidance from circulating blood cells (Figure 1). The identification of heterotypic CTC clusters supports this possibility. Interaction between tumor detached cells and macrophages can induce local inflammation that favors local extracellular matrix remodeling and CTC cluster migration ([17] for review). This tight crosstalk between macrophages and CTCs can even result into cell fusion (hybrid CTCs) that enhances invasiveness in glioblastoma [18]. In the same way, single CTCs were recently found to interact with white blood cells, mainly neutrophils. This interaction transcriptionally programs CTCs to establish junctions with neutrophils and promotes metastasis [19]. In addition, neutrophils generate neutrophil extracellular traps (NETs) also known as NETosis process that promotes extravasation of CTCs [20]. Alternative to white blood cell-mediated extravasation, CTCs, especially in clusters, might produce dynamic actin-rich protrusions of the plasma membrane, called invadopodia, known to promote the metastatic process. This is supported by the fact that isolated breast cancer single cell CTCs could grow in vitro as CTC tumor spheres, which generated invadopodia capable to become motile and to invade extracellular matrix [21]. This observation suggests that these specialized cytoskeleton structures known to boost invasiveness are probably involved in the detachment of CTC clusters and their trans-endothelial migration. This point needs to be explored.

Once in the circulation, single CTCs have been extensively described to setup interaction with platelets [17]. Association of CTCs in clusters appears to confer CTCs many advantages (Figure 1). In particular, as soon as the cells detach from the primary tumor, their keeping adhesion complexes fully assembled, in particular desmosomes, makes them resistant to anoïkis [10,22]. Interestingly, clustering may also down-regulate proliferation within the clusters, allowing the cells to escape treatments in small cell lung cancer [23]. However contradictory studies (for example, in Reference [24], in breast cancer) show that KI67, a marker of active proliferation, is found in CTC clusters, suggesting maintained proliferation. CTC resistance to cell death in CTC clusters might also be driven by depletion of genes involved in immune surveillance [10]. Finally, clusters might confer CTCs resistance to shear forces of the blood flow. 

## 3. CTC Cluster Molecular Features

### 3.1. Core Machinery Adhesion Proteins

CTC cluster biology is quite new, but the molecular basis of their specific features is described in several studies (Figure 2). Recent work has elegantly examined molecular profiling of CTC clusters in regards with single CTCs. In particular, CTC clusters predominantly express cell–cell adhesion proteins including those involved in tight junctions and desmosomes [10,22]. Single cell analyses found the desmosomal constituent plakoglobin encoded by the junction plakoglobin JUP gene, overexpressed by 219 folds in CTC clusters as compared to single CTCs from breast cancers [22]. The same group demonstrated that down-regulation of plakoglobin by shRNA in a mouse model of metastatic breast cancer resulted in drastic decrease of CTC clusters formation, whereas single CTC numbers remained unaffected. In the same way, overexpression of plakoglobin was found in vitro or in in vivo models of metastatic breast cancers, and its downregulation disrupted CTC clusters [25]. At the clinical level, plakoglobin expression in CTC clusters was an independent prognostic factor in patients with breast cancer [26]. In addition to plakoglobin, the glycoprotein CD44 might be involved in CTC clustering [13], but further validation on patient isolated CTC clusters is needed.

### 3.2. Stemness

Importantly, several studies agree on the fact that CTC clustering confers stemness properties to the cells. In particular, CD44, a cell surface marker often up-regulated in cancer stem cells (CSC), was overexpressed in CTC clusters. CD44 is involved in the maintenance of stemness signaling of numerous tumor cell types [27]. Similarly, stemness-associated genes were found up-regulated in breast cancer PDX-derived and patient-isolated CTC clusters [6]. This differential control of gene expression might be the result of differential methylation of genes encoding the stem cell specific transcription factors OCT4, NANOG, and SOX2, for example, in Reference [24]. In the same way, organoids derived from colorectal tumors released CTC clusters overexpressing CD44 [28]. Clustering of CTCs probably offers CSCs a favorable maintenance niche and the protection needed for their journey to distant metastatic sites. This is in agreement with the higher expression of metastatic niche modeling genes found in CTC clusters [10]. As CSCs are thought to be the main actors of metastatic dissemination [29], their traveling in CTC clusters might explain the recurrent link between metastatic disease and CTC clusters [10,22,26,30] (Figure 2).

### 3.3. Survival Advantage

The maintenance of cell-cell adhesion in CTC clusters might confer resistance to cell detachment-induced apoptosis called anoikis. This might be due to the overexpression of the anti-apoptotic Bcl2 protein found in CTC clusters as compared to single CTCs of breast cancer origin [6]. In the same way, the whole apoptotic pathway was found down-regulated by RNAseq profiling of CTC clusters produced by breast cancer PDXs. Interestingly, enrichment of gene groups related to cellular proliferation was reported in CTC clusters obtained from both PDX and patient samples [24]. Likewise, an increased percentage of KI67 positive cells was observed in CTC clusters as compared to single CTCs. Thus, the combination of apoptotic resistance and increased proliferation could explain the survival advantage of CTC clusters (Figure 2).

It is critical for CTCs to resist the shear stress induced by the physiological blood flow. This might in part be achieved by EMT plasticity carried by CTCs in clusters [31,32] and is in agreement to high expression of EMT markers in CTC clusters from colorectal spheroid tumor derived cell line [28]. This point is, however, challenged by the fact that no differences were found in the EMT signature between single CTCs and CTCS clusters [6]. Nevertheless, it is generally accepted that EMT is mandatory for high migratory capacity acquisition of single cells but seems less crucial for CTCs in clusters (see Reference [33,34] for review). Thus, the exact contribution of EMT in CTC clusters formation remains unclear. 

### 3.4. Immune System Escape

CTCs carry the remarkable capacity to escape immune system. Multiple mechanisms are involved in immune escape of CTCs in blood circulation and metastatic sites. For example, the presentation of surface immune checkpoint proteins is enhanced in CTCs [35] or antigen presentation processes are impaired (see Reference [36] for review). Noticeably, multiple pathways involved in immune response, such as type II interferon and TNF signaling pathways or antigen processing and presentation, are down-regulated in CTCs clusters as compared to single CTCs [6] (Figure 2), suggesting that they might be even more resistant. In the same way, Cheung et al. observed downregulation of MHC class II and T-cell activation genes [10]. Comparably, higher expression of the immune inhibitor CD24 [37] was observed in colorectal cancer-derived CTC clusters [28].

## 4. CTC Clusters in Clinic

### 4.1. Available Studies

Unlike single CTCs, CTC cluster detection and enumeration has not been intensely studied in the perspective of their clinical relevance. This might in part be due to the CTC detection methods that can damage clusters and limit their identification [38,39]. The vast majority of cancers demonstrating CTC clusters are solid cancers for 16 to 75% of patients (Table 1). This variability is cancer-dependent since CTC clusters in breast cancer ranged from 16 to 25% of patients or from 25 to 53% for lung cancer, while in pancreatic ductal adenocarcinoma, numbers raised up to 81%, but with important heterogeneity (from 18 to 81%, Table 1). In addition, definition of CTC clusters varies among the published data, two or more cells for some authors [40,41] to unfavorable clusters of more than 70 cells for others [42].

### 4.2. Prognostic Value

Although available studies reporting the link between CTC cluster identification and patient outcome are not numerous, it is possible to find correlations between the presence of CTC clusters and progression free survival for a majority of patients. Higher numbers of clusters are associated with shorter PFS underlining the possible link of clusters with the metastatic disease [28,38,39,40,41,42] (see Table 1 for examples in breast, pancreatic, lung, ovarian, prostate, and colorectal cancers). Results for the prognosis value of CTC clusters and overall survival are more contrasted. It is likely that, for patients already diagnosed with metastatic disease, CTC clusters might be systematically isolated, but their presence might not be examined in light of overall survival, unlike CTC numbers. They might even be counted as multi-numerous CTCs. A few studies, including ours, have reported that the presence of CTC clusters ( ≥ 1 cluster) was pejorative for patient survival for breast, pancreatic, prostate, colon, and lung cancers [30,40,43,44,45] without signs of metastatic disease at the time of analysis. In connection with these observations, a recent study focused on the prognostic value of in vitro breast and lung cancer CTCs cluster formation [46]. CTCs isolated from blood samples at baseline and during chemotherapy were cultured, and a correlation between CTC cluster phenotype and both therapeutic response and overall survival was found. This suggests that it is now time to switch from CTC hegemony to CTC cluster as a distinct entity with proper clinical value. Moreover, a few publications report the presence of CTC clusters in non-epithelial tumors, like glioblastoma, but their clinical significance was not assessed [47]. Thus, interest of CTC clusters enumeration and characterization is not limited to adenocarcinomas.

### 4.3. Improving Routine CTC Cluster Detection

Search of literature shows that CTC cluster identification is not systematically carried out when CTCs are counted in liquid biopsies. It is possible that the methods detecting CTCs destroy the clusters. In particular, methods with visible cells at the end point are more relevant for their identification than methods based on CTC enrichment followed by molecular identification at once. Another pitfall in CTC cluster detection might be the sampling site. A vast majority of studies search for CTCs in the peripheral blood, mostly by puncture of the cephalic vein. It is now more and more admitted that success of CTC detection, and more importantly CTC clusters might increase if samples are drawn in the immediate neighborhood of the primary tumor. This was confirmed by recent works [51,53], including ours [44]. In addition, microfluidic-based CTC isolation developed lately might contribute to increase CTC cluster identification rate and allow for their better characterization. In this line, precision medicine might benefit from individual cell study within clusters because increasing evidence supports their ability to reflect primary tumor heterogeneity, in particular, when assessing CTC clusters in the pulmonary vein, which were present in 50% of the patients [53].

## 5. Future Developments for CTC Clusters

### 5.1. Towards a Better Characterization

Taken together, published results suggest that CTCs and CTC clusters are not fundamentally different in terms of origin and global phenotype, except for their cell-cell adhesion capacities. Comparisons of single CTCs and CTC clusters have been carried out by transcriptomic and methylomic analyses. It would be interesting to assess their global proteomic and metabolomic to confirm their distinct molecular packages. It appears obvious that clustering enhances generic CTC properties in particular the metastatic potential and stemness. Interestingly, single cell analysis within CTC clusters could examine whether tumor cell migration as a cluster might be as determinant for metastatic properties and tumor aggressiveness as the generally accepted potentiation by genetic differences [56].

Both homotypic and heterotypic CTC clusters were evidenced from epithelial-derived solid tumors, as exemplified recently in a cohort of 70 patients with metastatic breast cancer (CTC clusters, 8.6%, and CTC–white blood cell clusters, 3.4% [19]). In blood cancers, few reports are available. However, in lymphomas models, heterotypic clusters with the CTC associated to stromal cells were more frequent than the homotypic ones [57]. Nevertheless, when coexisting, it would be very interesting to explore whether functional difference/resistance exist between homotypic clusters and heterotypic clusters comprising a unique CTC associated with a white blood cell or with platelets. 

Platelets are thought to confer resistance to cell death, EMT promotion, immune system escape, and extravasation. Thus, their association to CTCs promote their metastatic potential. Platelet association with CTC clusters was rarely reported [58], possibly because platelet-covered circulating tumor cells (CTC) are extremely difficult to isolate due to masking or down-regulation of surface epitopes [59]. Future studies should aim at exploring interactions between CTCs in clusters and platelets, since CTC clusters have been shown to overexpress platelet activation pathways [24].

Many clinical studies enumerated and analyzed isolated CTCs leaving aside CTC clusters, which are either not observed or less numerous than CTCs. CTC isolation techniques are in general not compatible with CTC clusters isolation. They exploit immunological and physical properties of CTCs (for review, see Reference [60]). In recent years, however, multiple isolation platforms suitable for both single CTCs and CTC clusters have been developed. A tridimensional scaffold chip coated with thermosensitive gelatin allowed the capture and retention of CTCs and CTCs clusters. Then, thermic dissolution of gelatin at 37 °C led to a gentle release of viable CTCs and CTC clusters [61]. Alternatively, microfluidic platforms can specifically isolate CTC clusters, including heterotypic, based on their physical properties (size) without any labeling and preserving their integrity [39,62]. Future developments should aim towards methods that allow phenotypic, molecular, and even functional analyses at once [63]. 

It is now well-admitted that intra-tumor heterogeneity conditions response to treatments. Accessibility to primary tumor is often challenging and invasive. Alternatively, single CTCs inform tumor diversity and treatment resistance. For example, global CTC heterogeneity measured by Shannon index can be used to adapt treatment in prostate cancer [64]. Whether CTC clusters carry the primary tumor heterogeneity deserves in-depth analysis, especially in light of the recent works showing oligoclonal nature of CTC clusters [10,22]. This is in agreement with the fact that CTC clusters derived from turnover models are more likely to contain virulent mutations [56], giving more chances to identify actionable mutations and access to personalized medicine. In fact, CTC clusters might become a more reliable tool to reflect primary tumor as compared to tissue biopsy and single CTCs.

### 5.2. Towards Innovative Treatment Leverage

Clinical studies show that CTC clusters arise mainly from solid tumors of epithelial origin, and are pretty rare and even rarer in early stage cancers. CTC cluster definition varies among studies and may be a source of heterogeneity of the clinical relevance. Variability of the methods employed to detect the clusters adds complexity to global interpretation of clinical data. Although mentioned as being observed in most studies, the number of CTC clusters as a separate entity from CTCs is not examined often enough. Allowing higher rates of purification with structural and viability preservation of CTC clusters, microfluidic plateforms appear like the best suited ones for this purpose. Future studies should focus on CTC cluster enumeration and its significance for the outcome of patients with metastatic disease, but also of early stages patients (Figure 3), as exemplified in lung [53] and pancreatic cancers [44]. In these latter cases, liquid biopsy performed in the vicinity of the tumor might increase chances to capture them [63,64,65].

The link between CTC clusters and resistance to treatments is becoming more and more obvious. Lee et al. reported platinium resistance in primary or recurrent ovarian cancer associated with the presence of CTC clusters in peripheral blood [55]. Although not explained by the authors, the study highlights a correlation between CTC clusters and metastatic evolution. In addition, resistance might result in higher viability in clusters than single CTCs when exposed to doxorubicin, as demonstrated by a multiple drug sensitivity assay of CTC or CTC clusters embedded in droplets [66]. CTC clusters could be predictors, as well as actors, of resistance. Association of CTC clusters with stemness, metastasis, and resistance to treatments represents key targetable vulnerabilities for cancer progression and/or relapse. In the same way, link between CTC clusters and poor disease prognosis suggests that inhibition of CTC cluster formation and release could reduce cancer dissemination. In particular, their elimination by cytapheresis might help kill resistance and dissemination [67]. Another promising therapeutic approach would disrupt clusters in transit. Focal Adhesion Kinase FAK inhibitor 14, a selective small molecular inhibitor of focal adhesion kinase or paclitaxel, a common chemotherapy, weakened cell–cell adhesion of CTC clusters and inhibited their migration [15]. In the same way, high scale testing of drugs (2486 FDA-approved compounds) identified the inhibitor of Na+/K+-ATPase ouabaïn and the tubulin binding vincristine as cluster disrupting agents, diminishing metastasis in a preclinical model [24].

In conclusion, more work needs to be done to determine if enhanced capacities of CTC clusters are causal of clustering or simply associative, as well as to consider them as promising targets for therapy. Of note though, they may already provide beneficial information for clinicians for diagnostic and prognostic purposes.

## Figures and Tables

**Figure 1 ijms-21-02653-f001:**
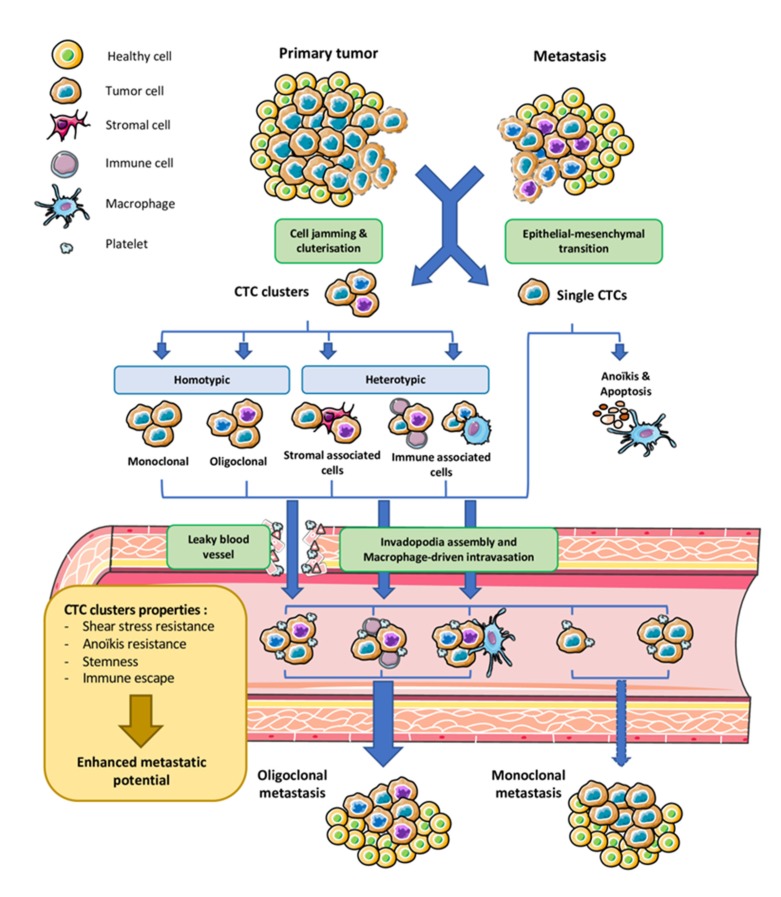
Origin and dissemination of circulating tumor cell (CTC) clusters. Cell aggregates detach from the primary tumor site and metastases by cell jamming to produce homotypic monoclonal or polyclonal tumor clusters. Released cells can also interact with stromal or immune cells in the inflammatory peri-tumoral infiltrate forming heterotypic clusters. Heterotypic cluster formation can also occur in blood vessels by association with circulating immune cells, and possibly with platelets. Intravasation of CTC clusters can occur by invadopodia and macrophage lead or through leaky blood vessels common in tumor microenvironment. In the bloodstream, clustering strengthens CTCs by anoïkis resistance, shear stress resistance, and immune escape and enhances their stemness, resulting in boosted metastatic potential. After extravasation in tissue with favorable microenvironment conditions, clusters can form monoclonal or polyclonal metastasis depending on their initial nature. This figure was performed using free online Servier Medical Art at www.servier.com.

**Figure 2 ijms-21-02653-f002:**
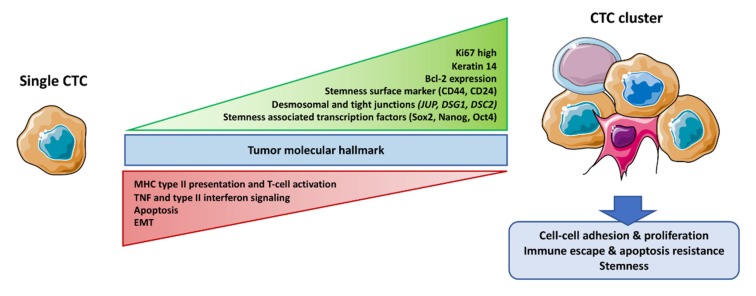
Molecular differences between CTC clusters and single CTCs. CTC clusters like single CTCs harbor the molecular hallmark of their primary tumor. By contrast, multiple pathways like cell–cell adhesion (desmosomes), stemness (surface markers and associated transcription factors), and proliferation (higher KI67) are up-regulated in CTC clusters. On the other hand, apoptosis and immune activation pathways like Major Histocompatibility Complex MHC type II presentation, T-cell activation, and Tumor Necrosis Factor TNF signaling are down-regulated. Epithelial to mesenchymal transition status of CTC clusters is crucial for single CTC migratory property but is debated for clusters. This figure was performed using free online Servier Medical Art at www.servier.com.

**Figure 3 ijms-21-02653-f003:**
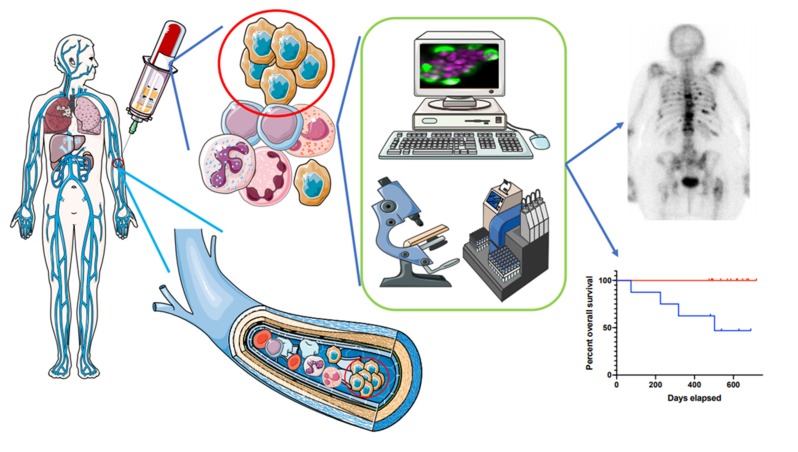
CTC clusters in clinical perspective. In the course of patient management, CTC cluster enumeration individualized from CTC detection might provide additional noninvasive information on patient prognosis, and might be a good companion biomarker of metastatic disease assessment. Further individual characterization of cells within CTC clusters will be helpful to measure intra-tumor heterogeneity and drug sensitivity/resistance. This figure was performed using free online Servier Medical Art at www.servier.com.

**Table 1 ijms-21-02653-t001:** Association of CTC clusters and clinical outcomes in cancer patients.

Patient Number	Stage: Localized, Metastatic, all Stage	CTC Enrichment /Isolation Method	CTC Identification/Detection Method	CTC Clusters Detection Rate in Cancer Patient (%)	Prognosis Value of CTC-Cluster Positive Detection	References
**Breast Cancer**						
52	Metastatic	CellSearch^®^	ICC: CK+, DAPI+, CD45−	8/32 (25%)	Correlate with shorter PFS	[41]
115	Advanced patient (Stage III/IV)	CellSearch^®^	ICC: CK+, DAPI+, CD45−	20/115 (17%)	Correlate with shorter PFS	[40]
52	Metastatic	CellSearch^®^	ICC: CK+, DAPI+, CD45−	9/52 (18%)	Correlate with shorter PFS and OS	[48]
128	Metastatic	CellSearch^®^	ICC: CK+, DAPI+, CD45−	21/128 (16%)	Correlate with shorter PFS and OS	[49]
156	Metastatic	CellSearch^®^	ICC: CK+, DAPI+, CD45−	30/156 (19%)	Correlate with shorter PFS and OS	[50]
Pancreatic ductal adenocarcinoma						
63	All stage	Microfluidic	ICC: CK+, DAPI+, CD45−	51/63 (81%)	Correlate with shorter PFS and OS	[42]
14	All stage	CellSearch^®^	ICC: CK+, DAPI+, CD45−	6/14 (42%) (in portal vein)	Not studied	[51]
20	Metastatic	Size-based: Screencell©	Cytology	13/20 (65%) before chemotherapy 15/20 (75%) after chemotherapy	No correlation between PFS and OS with cluster detection	[52]
22	Resectable	CellSearch^®^	ICC: CK+, DAPI+, CD45−	2/11 (18%) (portal blood)	Correlate with shorter OS	[44]
Lung cancer						
SCLC 97	All stage	CellSearch^®^	ICC: CK+, DAPI+, CD45−	25/97 (25%)	Correlate with shorter PFS and OS	[23]
NSCLC 36	Resectable disease (stage I-III)	OncoBean	ICC: CK+, DAPI+, CD45−	19/36 (53%)	Correlate with shorter PFS	[53]
NSCLC 29	All stage	Size-Based	ICC: CK+, DAPI+, CD45−	12/29 (41%)	Correlate with disease stage *	[54]
NSCLC 77	Resectable disease (stage I-III)	Size-based: Screencell©	Cytology	19/36 (52%) (13/19 in pulmonary vein)	Correlate with shorter PFS and OS	[30]
Epithelial ovarian cancer						
54	All stage	Microfluidic	ICC: CK+, DAPI+, CD45−	20/32 (62%)	Platinium resistance Correlate with shorter PFS	[55]
Prostate cancer						
98	All stage	Microfluidic	Flow-cytometry	49/98 (50%)	Correlate with shorter PFS and OS	[43]
98	All stage	Size based ISET	ICC: CK+, DAPI+, CD45−	18/98 (18%)	Correlate with shorter OS	[45]

OS, overall survival; PFS, progression free survival; ICC, immuno-cyto-chemistry; DAPI, 4′, 6-diamidino-2-phénylindole; CK, cytokeratin, a marker of epithelial cells; CD45, cluster of differentiation 45, a marker of white blood cells; CTCs: circulating tumor cells; ISET, isolation by size of epithelial tumor cells; NSCLC, non-small cell lung cancer; SCLC, small cell lung cancer; * Not statistically significant.

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
