# Peer review of "Circulating Tumor Cell Clusters: United We Stand Divided We Fall"

_ijms, 2020, doi:10.3390/ijms21072653_

Round 1
Reviewer 1 Report
Samuel et al. reviewed circulating tumor cell clusters in terms of its origin, molecular features and associated biological significance, clinical relevance as well as the future perspectives of clinical treatments of cancers. The descriptions are comprehensive and convincing, enough literature is included and cited although papers published after year of 2000 are strongly suggested. Overall, this review is relatively full-scale summarization of circulating tumor cell clusters in cancer development and suitable to be published.
Author Response
Please find the attached .docx file including our point by point answer to your comments

Reviewer 2 Report
The manuscript entitled “Circulating Tumor Cell Clusters: United we stand divided we fall” focused on origin, dissemination, and molecular features of circulating tumor cell (CTC) clusters. Although the abstract describes the importance of CTC clusters, the text related to “CTC clusters: united we stand divided we fall Over all” has not been explained, which is important for the readers those who are not working in this field – the authors need to include related explanation in the abstract. Are there any proteomic studies or other omics studies that have demonstrated the molecular differences between CTCs and CTC clusters? If available, including references related to the omics studies would improve the manuscript quality. In abstract, it has been stated “We will also consider their clinical relevance as prognostic biomarkers”. Emphasizing the text related to this statement in manuscript is important. The origin and dissemination has been depicted in Figure 1 with details. The molecular differences between CTCs and CTC clusters have been well presented in figure 2. The authors have been cited recent references. Overall, the manuscript has been well written and well presented the importance of circulating tumor cell clusters.
Author Response
Please find the attached .docx file including our point by point answer to your comments.

Reviewer 3 Report
The manuscript is interesting. There are some critical points that need attention.
- What different kinds of pathways are involved to make CTC? The author can collect the literature to elaborate properly. The authors may take an idea from the recent interesting paper where Geethadevi et al discussed the role of ERBB receptors on CTC formation of ovarian cancer and glioblastoma (Genes Cancer. 2017 PMID: 29321816; ERBB signaling in CTCs of ovarian cancer and glioblastoma).
- Please make a schema for different signaling that involved in circulating tumor cells formation.
- Please make a list with references for a List of small molecule inhibitors/Antibodies that inhibit CTC or are under clinical trial.
- Table 1. Association of CTC clusters and clinical outcomes in cancer patients. Did the author search CTC clusters and clinical outcomes for different cancer including Ovarian cancer and Glioblastoma?
- What are the detection techniques to detect CTS at clinical level? Please discuss.
Author Response

(The authors gave the same response as above.)

Round 2
Reviewer 3 Report
The authors have justified all the comments carefully. I have no further comments.